# Genome replication in asynchronously growing microbial populations

**Florian G. Pflug**[1], **Deepak Bhat**[2], **Simone Pigolotti**[1]*

**1** Biological Complexity Unit, Okinawa Institute of Science and Technology Graduate University, Onna, Okinawa, Japan, **2** Department of Physics, School of Advanced Sciences, Vellore Institute of Technology, Vellore, Tamil Nadu, India

\* simone.pigolotti@oist.jp

**Data Availability Statement:** All relevant data are within the manuscript and its Supporting information files. Code is available on GitHub: https://github.com/Biological-Complexity-Unit/dnareplication_PCB.

## Abstract

Biological cells replicate their genomes in a well-planned manner. The DNA replication program of an organism determines the timing at which different genomic regions are replicated, with fundamental consequences for cell homeostasis and genome stability. In a growing cell culture, genomic regions that are replicated early should be more abundant than regions that are replicated late. This abundance pattern can be experimentally measured using deep sequencing. However, a general quantitative theory linking this pattern to the replication program is still lacking. In this paper, we predict the abundance of DNA fragments in asynchronously growing cultures from any given stochastic model of the DNA replication program. As key examples, we present stochastic models of the DNA replication programs in budding yeast and *Escherichia coli*. In both cases, our model results are in excellent agreement with experimental data and permit to infer key information about the replication program. In particular, our method is able to infer the locations of known replication origins in budding yeast with high accuracy. These examples demonstrate that our method can provide insight into a broad range of organisms, from bacteria to eukaryotes.

## Author summary

Biological cells replicate their genome in a planned manner. One way of obtaining experimental information about this plan is by deep sequencing a growing culture of cells. The idea underlying these experiments is that genomic regions that are replicated earlier would be present in higher abundances than regions replicated at a later stage. In this paper, we make use of this idea to obtain precise quantitative information on replication programs from sequencing. As a main application, we infer the locations of origins of replication in budding yeast from sequencing data. Our inference is consistent with direct experimental evidences on the locations of these origins. Our method can be in principle be applied to any organism that can be cultured and sequenced, and has therefore the potential to shed light on the replication program of a broad class of organisms.

**Funding:** This work was supported by JSPS KAKENHI Grant No. 23H01146 (to SP). DB thanks Vellore Institute of Technology, Vellore for providing VIT SEED Grant-RGEMS Fund (SG20220060) for carrying out this research work. The funders had no role in study design, data collection and analysis, decision to publish, or preparation of the manuscript.

**Competing interests:** The authors have declared that no competing interests exist.

## Introduction

The genome of an organism contains precious information about its functioning. Genomes need to be reliably and quickly replicated for cells to pass biological information to the next generation. Replication of a genome proceeds according to a certain plan, termed the "replication program" [1–4]. For example, most bacteria have a circular genome, where two replisomes initiate replication by binding at the same origin site [5]. They replicate the genome in opposite directions. Replication is completed when they meet, after each of them has copied approximately half of the genome. The replication program is carefully orchestrated, but not completely deterministic. Stochasticity is particularly relevant in eukaryotes and archaea, where many origin sites are present in each chromosome [6, 7]. Replication can initiate at these origin sites at different times. These times are characterized by some degree of randomness [8, 9] and often only a subset of origins are activated at all [1].

Replication programs must be coordinated with the cell cycle in some way. In eukaryotes, replication takes place at a well-defined stage of the cell cycle (the S phase), during which different genomic regions are replicated at different times [8]. In bacteria, replication initiation is carefully timed relative to the cell cycle as well [10–12]. The interplay between the replication program and the cell cycle is crucial when trying to infer the replication program from experimental observations. Many experiments study samples in which all cells are approximately at the same stage of the cell cycle. This can be achieved by either arresting the cell cycle at a certain stage or by cell sorting [4]. The fraction of these synchronized cells that have copied each genome location can be then measured using deep sequencing. This approach has been extensively used to study the eukaryotic replication program [4, 13, 14].

An alternative method is to measure the abundance of DNA fragments in asynchronous, exponentially growing populations. This approach, traditionally called marker frequency analysis [15], has been extensively applied to bacterial replication, for example to study *Escherichia coli* mutant lacking genes that assist DNA replication [16–18] and artificially engineered *E.coli* strains with multiple replication origins [19]. The asynchronous approach is experimentally much simpler and avoids potential artifacts caused by the cell cycle arrest or cell sorting [4]. Progress in DNA sequencing have made these experiments high-throughput and relatively inexpensive. However, the DNA sampled in these experiments originates from a mixture of cells at different stages in their life cycle, rendering the theoretical interpretation of such data problematic [20].

Theoretical approaches have attempted to describe measurements in asynchronous populations. However, these approaches either neglect stochasticity, or are limited to specific model systems. For example, we recently proposed a stochastic model describing DNA replication in growing *E.coli* populations [21]. A broader range of bacterial systems have been studied assuming a deterministic replication program [22, 23]. Finally, a model of the replication program in budding yeast adopted the working hypothesis that cells in an asynchronous population are at random, uniformly distributed stages in their cell cycle [3].

In this paper, we develop a general theory to infer the replication program from sequencing of an asynchronously growing population of cells. Our theory builds upon classic results on age-structured populations [15, 24–27], that we extend to populations of stochastically replicating genomes. Our approach requires minimal assumptions on the replication dynamics. In particular, it allows for a stochastic replication program and equally applies to bacteria and eukaryotes. We apply our method to models of the DNA replication program in budding yeast and *E. coli*. We provide exact solutions for both of these models. These solutions fit existing experimental data very well. In the case of yeast, our approach permits us to reliably infer the location of replication origins. In the case of *E. coli*, our model sheds light on recently observed oscillations of bacterial replisome speed.

## Methods

### General theory

We consider a large, growing population of cells. We call $N_c(t)$ the number of cells that are present in the population at time $t$. Each cell may contain multiple genomes, some complete and other undergoing synthesis (incomplete). We denote by $N_g(t)$ the the total number of complete and incomplete genomes in the population.

Our theory is based on the following assumptions. The number of cells grows exponentially: $N_c(t) \propto \exp(\Lambda t)$, where $\Lambda$ is the exponential growth rate. The population grows in a steady, asynchronous manner. This assumption implies that the average number of genomes per cell must remain constant, and therefore $N_g(t) \propto \exp(\Lambda t)$. Genomes in the population are immortal, i.e. we neglect the rate at which they might be degraded. All genomes in the population are statistically identical, in the sense that they are all characterized by the same stochastic replication program. These assumptions are realistic in common experimental situations. Moreover, some of them can be readily relaxed, if necessary (see S1A Appendix).

We now assign an age to each genome in the population. To this aim, we conventionally set the birth time of a daughter genome at the start of the replication process that generates it from a parent genome, see Fig 1a. We note that the distinction between a "parent" and a "daughter" genome is somewhat arbitrary, as each of them is made up of a preexisting strand and a newly copied complementary one. Since genomes are immortal, the probability density of new genomes is proportional to $\dot{N}_g(t)$, which is proportional to $\exp(\Lambda t)$ as well. It follows that the distribution of ages $\tau$ of genomes in the population must be proportional to $\dot{N}_g(t - \tau)$. From this fact, we conclude that the distribution of genome ages in the population is

$$P(\tau) = \Lambda e^{-\Lambda \tau}. \tag{1}$$

We now look into the DNA replication program in more details. In bacteria, replication is carried out by a pair of replisomes, that bind at a specific genome site (the origin) and proceed in opposite directions, each replicating both strands. DNA replication is substantially more complex in eukaryotes, where a large number of replisomes replicate the same chromosome, and initiation sites might be stochastically activated. We encapsulate the outcome of all of these processes into the probability $f(x, \tau)$ that the genome location $x$ has already been copied in a genome of age $\tau$. By definition, $f(x, \tau)$ is a non-decreasing function of $\tau$. Our assumption that genomes are statistically identical means that all genomes are characterized by the same $f(x, \tau)$.

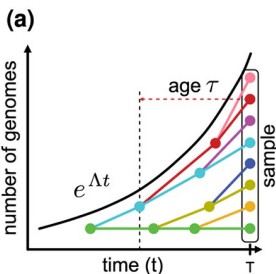
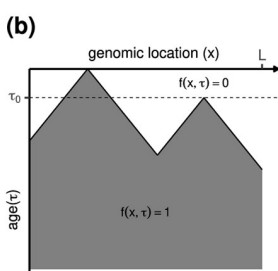
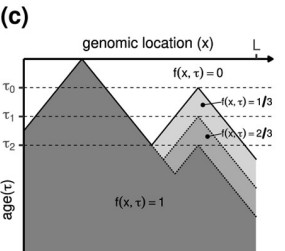
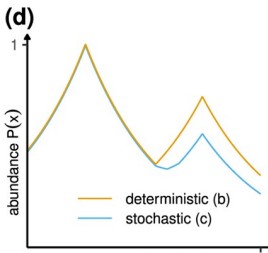

**Fig 1. DNA abundance in an asynchronous, exponentially growing microbial population. (a)**. Total number of genomes (black line) and genealogy of individual genomes (colored tree). Nodes in the tree represent replication initiations. Such events leave the template unchanged and create a new genome (differently colored descendant) with initial age $\tau = 0$. **(b)**. Example of a deterministic replication program $f(x, \tau)$ on a linear genome in which replication is initiated at two origins, one firing at age $\tau = 0$ and one at age $\tau = \tau_0$, and proceeds deterministically with constant speed. We recall that the function $f(x, \tau)$ represents the probability that location $x$ is replicated by time $\tau$. **(c)**. A stochastic version of the replication program in (b), in which the second origin fires randomly at either $\tau_0$, $\tau_1$ or $\tau_2$. **(d)**. DNA abundance distribution predicted by Eq (2) from replication programs (b) and (c).

Examples of deterministic and stochastic replication programs are represented in Fig 1b and 1c, respectively.

We now introduce the probability $\mathcal{P}(x)$ that a randomly chosen genome in the population contains the genome location $x$. According to our definition, incomplete genomes that are undergoing synthesis form part of the genome population, together with complete ones. This already suggests that $\mathcal{P}(x)$ must contain information about the size spectrum of incomplete genomes, which in turn depends on the replication program. We also remark that $\mathcal{P}(x)$ is not necessarily normalized to one when integrated over the entire genome. Its normalized counterpart represents the probability density that a randomly chosen genome fragment in the population originates from genomic location $x$. For this reason, we call $\mathcal{P}(x)$ the DNA abundance distribution. The DNA abundance distribution can be experimentally measured using deep sequencing.

By using Eq (1) and integrating over all genome ages, we find that $\mathcal{P}(x)$ is related with $f(x, \tau)$ by

$$\mathcal{P}(x) = \int_0^\infty d\tau \, \Lambda e^{-\Lambda \tau} f(x, \tau), \tag{2}$$

see Fig 1d. To find a more transparent expression, we introduce the probability density $\psi(x, \tau) = \partial_\tau f(x, \tau)$ of the time $\tau$ at which a particular location $x$ is replicated. Substituting this definition into Eq (2) and integrating by parts we obtain

$$\mathcal{P}(x) = \left\langle e^{-\Lambda \tau} \right\rangle_x, \tag{3}$$

where $\langle \ldots \rangle_x = \int_0^\infty d\tau \ldots \psi(x, \tau)$ is the average over the distribution of replication ages. Eq (2), or equivalently Eq (3), is the basis of our approach.

In our derivation, we intentionally avoided modeling the specific dynamics of the cell cycle, how it is regulated, and how it is coordinated with the DNA replication program [28–30]. Our theory is rigorously valid independently of these aspects, provided that our initial assumptions hold.

We also note that the absolute timing of initiation of the replication program can not be inferred in our framework. Indeed, it follows from Eq (3) that changing the initial time of replication (or its uncertainty) would alter $\mathcal{P}(x)$ by a multiplicative factor. This factor is not empirically measurable, because with deep sequencing we can only measure $\mathcal{P}(x)$ up to a normalization constant. However, in organisms with multiple origins, firing time lags between different origins would alter the relative abundance of different genomic regions and would therefore be measurable with our approach.

## Deterministic limit

In the simple case where replication proceeds deterministically and the replisome speed is a function of its position on the DNA, one has

$$\psi(x, \tau) = \delta \left( \tau - \tau_0 - \int_{x_0}^{x} \frac{dx'}{v(x')} \right) \tag{4}$$

where $v(x)$ is the replisome speed at position $x$, $x_0$ is the coordinate of the replication origin for the replisome that copied position $x$, and $\tau_0$ is the firing age for that origin. The speed $v$ might take positive or negative values depending on whether the replisome proceeds in the positive

genome direction. It follows from Eqs (3) and (4) that

$$\mathcal{P}(x) = \exp\left(-\Lambda\tau_0 - \Lambda \int_{x_0}^x \frac{dx'}{v(x')}\right). \tag{5}$$

Solving for the speed, we obtain

$$v(x) = -\Lambda \left[\frac{d}{dx}\ln\mathcal{P}(x)\right]^{-1}, \tag{6}$$

i.e., the replication speed is inversely proportional to the logarithmic slope of $\mathcal{P}(x)$ [22, 23]. An advantage of the deterministic assumption is that it leads to a one-to-one correspondence between DNA abundance and local replisome speed thanks to Eq (6). However, in many realistic cases, neglecting stochasticity might lead to inaccurate predictions.

Unfortunately, in the general stochastic case, different replication programs might give rise to the same DNA abundance distribution. This implies that one cannot directly invert Eq (3) to express $f(x, \tau)$ in terms of $\mathcal{P}(x)$. In these situations, one has to complement the information contained in $\mathcal{P}(x)$ with modeling assumptions, as exemplified in the following.

## Results

### Eukaryotic DNA replication

In eukaryotes, replication is initiated from many randomly activated replication origins. When an origin fires, two replisomes are formed and start moving from that origin in opposite directions. Origins are prevented from firing on stretches of DNA that have already been duplicated. To describe this process, theoretical progress [31, 32] has made use of an analogy with freezing/crystallization kinetics, as described by the so-called Kolmogorov-Johnson-Mehl-Avrami model [33–35], see also [36, 37]. We briefly summarize this idea and then extend it to asynchronously growing populations.

In principle, a given location $x$ on an eukaryotic genome of age $\tau$ could have been replicated by different replisomes starting from different origins. The replication program $f(x, \tau)$ can be seen as the probability that the past "light-cone" $V_{x,\tau}$ of the space-time point $x$, $\tau$ contains at least one origin firing event. The past light-cone $V_{x,\tau}$ represents the set of space-time points $x'$, $\tau'$ from which $x$ is reachable by a replisome within a time $\tau$, see Fig 2a. This elegant argument circumvents the problem of determining from which origin did the replisome that replicated the genome location $x$ started.

Assuming that replisomes progress deterministically with constant speed $v_0$, the past light-cone is expressed by

$$V_{x,\tau} = \{(x', \tau') \,|\, |x - x'| \le v_0(\tau - \tau')\}, \tag{7}$$

see Fig 2a. Following [1, 38], we now assume that origins attempt to fire independently from one another with stochastic rates $I(x, \tau)$. The probability that location $x$ has been replicated by age $\tau$ is then expressed by $f(x, \tau) = 1 - \exp(-\int\int_{V_{x,\tau}} dx' d\tau'\, I(x', \tau'))$, where we took into account that each origin can fire only once (Fig 2b). This expression would remain valid if we chose a different replisome dynamics, corresponding to a form of the light-cone $V_{x,\tau}$ other than that expressed by Eq (7). Using Eq (2), we express the DNA abundance distribution as

$$\mathcal{P}(x) = 1 - \Lambda \int_0^\infty d\tau \exp\left(-\Lambda\tau - \int\int_{V_{x,\tau}} dx' d\tau'\, I(x', \tau')\right). \tag{8}$$

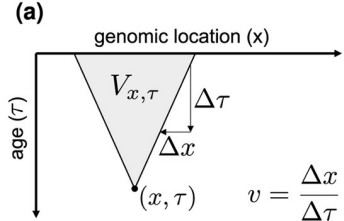
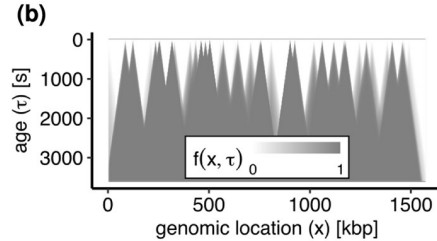
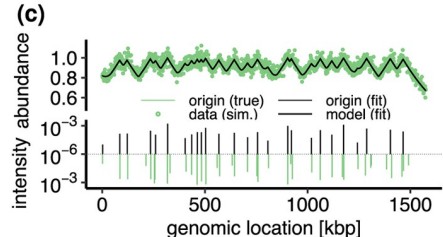

**Fig 2. Eukaryotic model. (a)**. Past light-cone $V_{x,\tau}$ of a space-time point $(x, \tau)$. At least one origin must have fired within the light cone for location $x$ to be replicated by time $\tau$. **(b)**. Eukaryotic replication program from Eq (9) for the annotated origins on *S. cerevisiae* W303 chromosome IV with intensities $I_i^\star/v$ randomly log-uniformly distributed on $[10^{-5}, 2 \cdot 10^{-3}]$. **(c)**. Inference of origin locations and intensities from a simulated DNA abundance. Top: the model (black line) fitted to simulated abundances (green line). Bottom: inferred 26 origins (black bars) and 39 true origins (green bars). Rescaled intensities $I_i^\star/v$ [1/bp] are plotted in log-scale. Parameters are: $v = 27$ bp/s, $\Lambda = 9.6 \cdot 10^{-5}$ 1/s (doubling time 120 minutes). The true origin locations and intensities are those from panel (b). The resulting true abundances are multiplied by a Gamma-distributed random variable with mean 1.0 and coefficient of variation 0.04 to mimic measurement errors.

We now focus on budding yeast, where origins correlate with specific DNA motifs and are therefore thought to be at well-defined locations $x_1, \ldots, x_K$ on the chromosomes [6]. We further assume that firing rates are constant in time, so that $I(x, \tau) = \sum_{j=1}^{K} I_j^\star \delta(x - x_j)$. We note that the origin firing rate was observed to be time-dependent in budding yeast [8]. This means that this second assumption is not fully accurate and should be taken as a simplifying approximation. Under these assumptions, the DNA abundance is given by

$$\mathcal{P}(x) = \sum_{k=1}^{K} \left( \frac{1}{W_{k-1}} - \frac{1}{W_k} \right) e^{-\Lambda \mathcal{T}_k} \quad \text{with}$$

$$\tau_k = \frac{|x - x_k|}{v_0}, \qquad \mathcal{T}_k = \tau_k + \sum_{j=1}^{k} \left( \tau_k - \tau_j \right) I_j^\star/\Lambda, \qquad W_k = 1 + \sum_{j=1}^{k} I_j^\star/\Lambda. \tag{9}$$

where $\tau_k$ is the travel time from the $k$-th origin to location $x$ and we ordered the origins, for each given $x$, such that $0 < \tau_1 < \cdots < \tau_K$. Eq (9) is derived in S1B Appendix.

We implemented a simulated annealing algorithm to infer the origin number, location, and intensities based on DNA abundance data via Eq (9). Our inference procedure treats $\Lambda/v_0$, the number of origins $K$, the origin positions $x_1, \ldots, x_K$, and the re-scaled firing rates $I_1^\star/v_0, \ldots, I_K^\star/v_0$ as free parameters. We call the compound parameter $I_j^\star/v_0$ the intensity of origin $j$. Our algorithm uses the Akaike information criterion (AIC) as the cost function to avoid over-fitting (details in S1C Appendix).

We first tested this inference procedure on artificial data, in which the genome length (1.6Mbp) and origins distribution are comparable to those of the longest chromosome of budding yeast (chromosome IV). Our inference algorithm detected the location and intensity of 26 out of 39 true origins with high accuracy (Fig 2c; median distance to true origin 1.5kb, median relative error of intensity 40% if non-resolved clusters of true origins are merged; true intensities range over two orders of magnitude). The 13 non-recovered origins either have low intensity or were merged with another origin in close proximity.

## Inferring yeast replication origins from experimental data

We now apply our model and inference procedure to experimentally measured DNA abundance in an asynchronous populations of budding yeast (*S. cerevisiae* W303) [39]. Our algorithm infers $K = 234$ origins across the 16 yeast chromosomes, which is about 30% less than

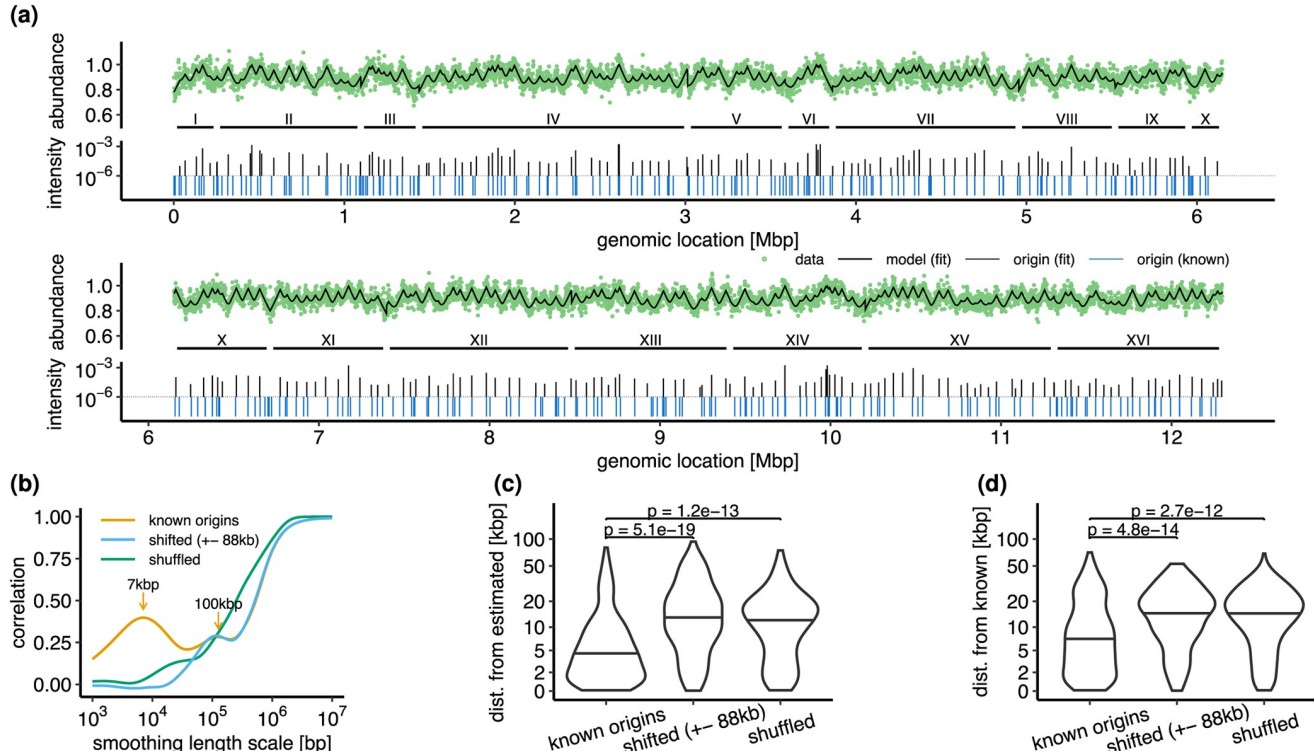

**Fig 3. Eukaryotic model fitted to *S. cerevisiae* data from [39].** Fitted parameters are: $\Lambda/v$, origin positions $x_1, \ldots, x_n$ and intensities $I_1^\star/v, \ldots, I_n^\star/v$. The fitted number of origins is $K = 234$. Known origins used for validation were lifted from the annotated genome for strain 288C (RefSeq assembly *R64-3-1*, accession *GCF_000146045.2*) using *liftoff* [40]. We excluded the mitochondrial genome. **(a)**. Observed (green circles) and predicted (black line) DNA abundances (top) and inferred origin positions and intensities (bottom). Known origin positions (blue) shown for comparison. **(b)**. Correlation of estimated densities of predicted ($K = 234$) and known ($K = 354$) origins at different smoothing length scales (yellow). Y-axis shows the Pearson correlation between densities of known and predicted origins computed using kernel density estimation (KDE) using a Gaussian kernel with variance equal to the indicated smoothing length scale. For comparison, we show the correlation with the density of the known origins after: (i) shifting each origin up to ±5× the average half-distance between origins (i.e. ± 88kb) using uniformly randomly generated displacements (blue). (ii) shuffling by re-arranging chromosomes I-XVI but keeping the relative positions within each chromosome the same (green). **(c)**. Distribution of distances between estimated and closest known origins. **(d)**. Distribution of distances between known and closest estimated origins. In (c) and (d), the y-axis is non-linearly scaled to enlarge the region of interest. Horizontal lines indicate the medians. Reported p-values are computed using Mann–Whitney U tests.

the number of known origins for *S. cerevisiae* W303 ($K = 354$). The predicted DNA abundance well matches the experimental one, see Fig 3a.

Our method well predicts origins of replication with a resolution on the order of a few kilobases, and without using additional experimental information other than the DNA abundance distribution. Comparing the densities of inferred and known origins of *S. cerevisiae* W303 at different length scales shows a peak at about 7kb. This length scale can be interpreted as the spatial resolution of the inferred origin locations, see Fig 3b. A second correlation peak at 100kb suggests a large-scale pattern in the distribution of origins. As expected, the peak at 7kb vanishes when known origin locations are randomly shifted by ±88kbp, or shuffled by randomly reordering chromosomes. The results of the correlation analysis are corroborated by matching each inferred origin to the closed known origin (Fig 3c; median distance 3.9kb) respectively each known origin to the closest inferred origin (Fig 3d; median distance 6.6kb). In both cases, the average distances are significantly increased if the known origins are shifted or shuffled.

## Bacterial DNA replication

In this Section, we introduce a stochastic model for the DNA replication of the bacterium *E. coli*. The model accounts for variations in the speed of replication, as recently observed [21]. In addition, replisomes in the model can stochastically stall, as observed in single molecule experiments [41, 42].

Most bacteria, including *E. coli*, have a single circular chromosome that is replicated by two replisomes. The two replisomes start from the same origin site, one proceeding clockwise and the other counterclockwise. Replication is completed when the two replisomes meet. We assume that the two replisomes do not backtrack and that they act independently from one another until they meet. A base is therefore replicated by whichever replisome reaches it first. The joint replication program of two replisomes is therefore $f(x, \tau) = 1 - [1 - f_1(x, \tau)][1 - f_2(x, \tau)]$, where $f_1(x, \tau)$ and $f_2(x, \tau)$ are the individual replication programs of each replisome, i.e. the probabilities that position $x$ has been replicated at age $\tau$ by the respective replisome. Substituting this expression in Eq (2) we obtain

$$\mathcal{P}(x) = 1 - \int_0^\infty d\tau\, \Lambda e^{-\Lambda\tau}(1 - f_1(x, \tau))(1 - f_2(x, \tau)). \tag{10}$$

We define the genome coordinate $x \in [0, L]$ where $L$ is the genome length. We set the coordinate of the replication origin at $x = 0$ (and equivalently $x = L$, since the genome is circular). We call $x_1(\tau)$ and $x_2(\tau)$ the positions of the two replisomes along this coordinate as a function of age. The first replisome starts at $x(0) = 0$ and moves in the direction of increasing $x$, whereas the second starts at $x(0) = L$, where $L$ is the genome length, and moves in the direction of decreasing $x$. We express the replisome dynamics in terms of two Langevin equations

$$\frac{d}{d\tau}y_1 = v_0 h(\tau) + \sqrt{2Dh(\tau)}\, \xi_1(\tau), \quad \frac{d}{d\tau}y_2 = -v_0 h(\tau) + \sqrt{2Dh(\tau)}\, \xi_2(\tau),$$
$$x_1(\tau) = \max_{\tau' \le \tau} y_1(\tau'), \qquad\qquad x_2(\tau) = \min_{\tau' \le \tau} y_2(\tau')\ , \tag{11}$$

where $\xi_1(\tau)$ and $\xi_2(\tau)$ are Gaussian white noise sources. The function $h(\tau)$ represents a temporal modulation of the replisome speed, as a consequence of varying conditions during the cell cycle [21]. We assume for convenience that the diffusivity is modulated by the same function. We also assume for simplicity that the two replisomes are equally affected by these fluctuations. Eq (11) can be interpreted as follows: Whenever $y_i(\tau)$ attains a new maximal distance from its origin, then $x_i(\tau)$ is moving forward and it coincides with $y_i(\tau)$. If, instead, $y_i(\tau)$ is making a negative excursion from its past maximal distance, then $x_i(\tau)$ stalls, i.e. remains frozen at the value of the last maximal distance of $y_i(\tau)$. The dimensionless Peclet number $\mathrm{Pe} = Lv_0/4D$ controls the commonness of long stalling events; for large Pe, the dynamics are nearly deterministic and long stalling events are rare (Fig 4a). Stochastic trajectories generated by the model are qualitatively similar of those observed in single-molecule experiments with DNA polymerases [41, 42]. We remark that our model describes stochastic, position-independent stalling, in contrast with more regular stalling at specific position as observed in *Bacillus subtilis* [22].

A consequence of Eq (11) is that the individual replication program $f_i(x, \tau)$ is equal to the first-passage probability of the associated process $y_i$ through $x$. This first-passage probability is expressed by the inverse Gaussian distribution

$$f_i(x, \tau) \quad = \int_0^{H(\tau)} \sqrt{\frac{\mu_i^3}{2\pi\sigma_i^2 u^3}} \exp\left(-\frac{1}{2}\frac{(u - \mu_i)^2}{\sigma_i^2}\frac{\mu_i}{u}\right) du\ , \tag{12}$$

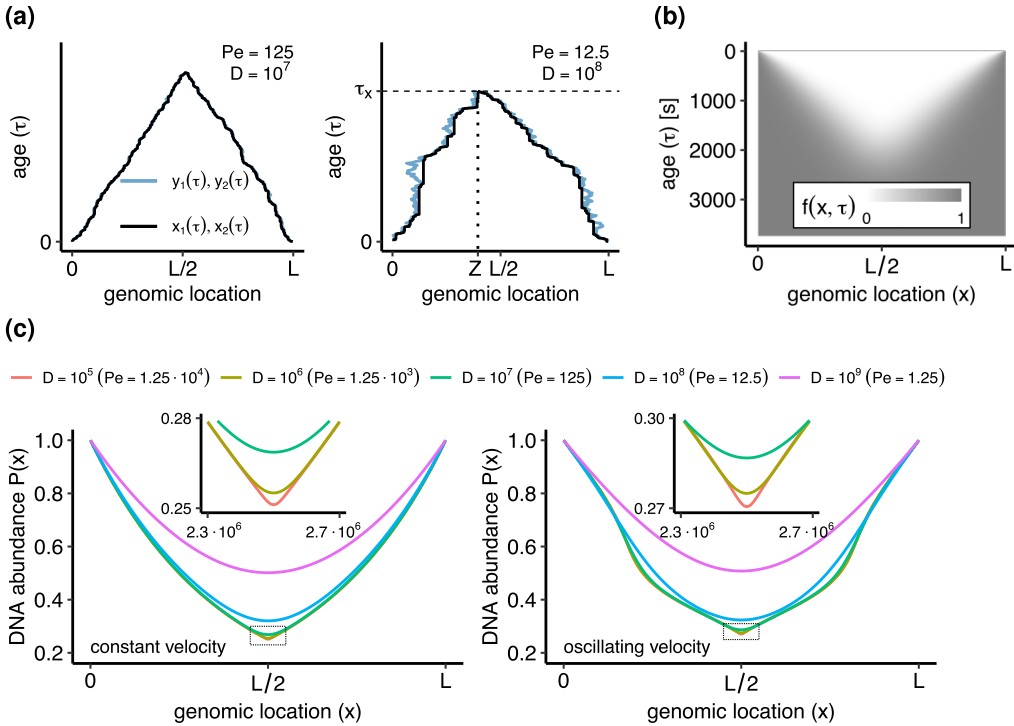

**Fig 4. Bacterial DNA replication model.** Parameters are: genome length $L = 5 \cdot 10^6$, growth rate $\Lambda = 2\mathrm{h}^{-1}$, baseline speed $v_0 = 10^3$ bp/s. **(a)**. Trajectories $x_1, x_2$ of the model (black lines) and auxiliary processes $y_1, y_2$ (blue lines). Backwards movements of $y_1, y_2$ correspond to stochastic stalling of $x_1, x_2$. Replication concludes at an age $\tau_x$ when $x_1$ and $x_2$ first meet at a random meeting point $Z = x_1(\tau_x) = x_2(\tau_x)$. **(b)**. Bacterial replication program $f(x, \tau) = 1 - (1 - f_1)(1 - f_2)$ with $f_1, f_2$ from Eq (12) for $D = 10^8$ bp$^2$/s. **(c)**. DNA abundance distribution for constant and oscillating replisome speed and different values of $D$. In the constant speed case, $h(\tau) = 1$, whereas in the oscillating speed case $h(\tau) = 1 + \delta \cos(\omega\tau + \phi)$ with $\delta = 0.5$, $\omega = 2\pi/1800$, and $\phi = 0$.

where $H(\tau) = \int_0^\tau h(u)du$ and

$$
\begin{aligned}
\mu_1 &= x/v_0, & \mu_2 &= (L - x)/v_0, \\
\sigma_1^2 &= 2Dx/v_0^3, & \sigma_2^2 &= 2D(L - x)/v_0^3 \ .
\end{aligned}
\tag{13}
$$

Eqs (12) and (13) are derived in S1D Appendix. We compute the DNA abundance $\mathcal{P}(x)$ by substituting Eq (12) into Eq (10), see Fig 4b. We numerically evaluate the final integral over $\tau$ appearing in Eq (10).

The main effect of diffusivity is to smooth the DNA abundance distribution around the expected meeting point $x = L/2$ of the two replisomes (see Fig 4c). Briefly, for $D = 0$ the DNA abundance exhibits a cusp, whereas for positive $D$ the abundance is smooth, see S1E Appendix. From the point of view of trajectories, this smoothing occurs because the two replisomes do not necessarily meet exactly at $x = L/2$. The uncertainty on the location $Z$ of the meeting point is approximately equal to

$$
(\Delta Z)^2 = \langle (Z - L/2)^2 \rangle \approx \frac{DL}{2v_0} = \frac{L^2}{8\mathrm{Pe}} \ .
\tag{14}
$$

Eq (14) is derived in S1F Appendix.

**Table 1. Parameter estimates for time-dependent speed $v(t) = v_0(1 + \delta \cos(\omega t + \phi))$.** For $v_0, D, \delta, \omega, \phi$ the reported standard errors represent the variability over replicates. The Peclet number $Pe = Lv_0/4D$ and meeting point uncertainty $\Delta Z = \sqrt{DL/2v_0}$, see Eq (14), are computed from the average estimates of $v_0$ and $D$ over replicates where $D > 0$. Their standard error are estimated using error propagation.

| $T$ [˚C] | $v_0$ [bp/s] | $D$ [kbp²/s] | $\delta$ | $\omega$ [rad/h] | $\phi$ [rad] | $Pe$ | $\Delta Z$ [kbp] |
|---|---|---|---|---|---|---|---|
| 17 | 230 ± 30 | 1.6 ± 1.4 | 0.29 ± 0.1 | 5.8 ± 4.2 | 1.3 ± 1.5 | 100 ± 100 | 130 ± 50 |
| 22 | 350 ± 21 | 0.5 ± 0.8 | 0.19 ± 0.07 | 3.9 ± 0.6 | 3.1 ± 0.5 | 200 ± 30 | 110 ± 7 |
| 27 | 530 ± 20 | 1.0 ± 1.2 | 0.19 ± 0.03 | 8.7 ± 0.4 | 2.0 ± 0.2 | 400 ± 400 | 80 ± 40 |
| 32 | 810 ± 57 | 1.7 ± 1.8 | 0.12 ± 0.04 | 16 ± 1.3 | 1.5 ± 0.2 | 500 ± 400 | 80 ± 30 |
| 37 | 950 ± 24 | 1.2 ± 2.3 | 0.19 ± 0.03 | 15 ± 0.4 | 2.9 ± 0.2 | 500 ± 600 | 70 ± 50 |

## Inferring speed fluctuations in E. coli from experimental data

We now fit our model of replisome dynamics to experimentally measured DNA abundance from wild type *E. coli* grown at different temperatures (from Ref. [21]). We assume that the speed and diffusion coefficient are modulated in time by a function

$$h(\tau) = 1 + \delta \cos(\omega \tau + \phi), \tag{15}$$

see Fig 4. In the fit, we treat $v_0$, $D$, $\delta$, $\omega$, and $\phi$ as free parameters (see S1H Appendix).

The model fits the experimental DNA abundances very well, see Fig 5a. The estimates of the mean speed $v_0$ are highly consistent among replicates across all temperatures, see Table 1. At temperatures above 17˚C we find robust evidence of speed fluctuations. The model provides consistent estimates of $\delta$, $\omega$ and $\phi$ in these cases, with an improvement of the quality of fit ranging between 20% to 40% percent compared to the constant-speed case (Fig 5b). At

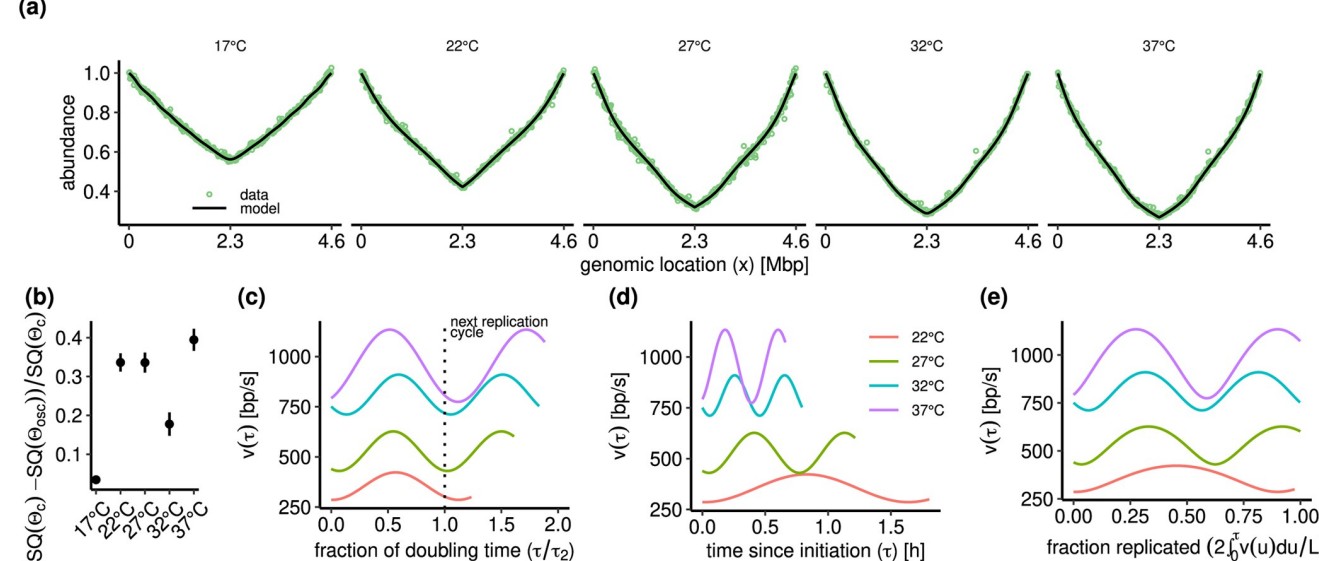

**Fig 5. Bacterial model fit to the data of Ref. [21].** The replisome speed is modulated by an oscillatory function, see Eq (15). We fitted the parameters $v_0$, $D$, $\delta$, $\omega$, and $\phi$ from the measured DNA abundances. The growth rate $\Lambda$ was independently measured in the experiments (see S1H Appendix). **(a).** Observed DNA abundance and model predictions for E. coli cultures growing at temperatures $T = 17$˚C, $22$˚C, $27$˚C, $32$˚C, $37$˚C. **(b).** Relative decrease in residuals ($SQ(\Theta) = \sum_{i=1}^{N} (a_i - \lambda\mathcal{P}(x_i|\Theta))^2/\sigma_i^2$, see S1H Appendix) of the model ($\Theta_{osc}$) vs. the constant speed case ($\Theta_c$). **(c)** Average instantaneous speed $v(\tau) = v_0 h(\tau)$ as a function of the fraction $\tau/\tau_2$ of the doubling time $\tau_2 = \log(2)/\Lambda$. **(d).** Average instantaneous speed $v(\tau) = v_0 h(\tau)$ as the function of the time $\tau$ since replication initiation.**(e).** Average instantaneous speed $v(\tau) = v_0 h(\tau)$ as a function of the replication progress, i.e. of the fraction $2\int_0^\tau v(u)du/L$ of replicated genome. In (c-e) we omitted 17˚C since the effect of speed fluctuations on $\mathcal{P}(x)$ is negligible at that temperature (see panel b).

17°C, the effect of the speed fluctuations on $\mathcal{P}(x)$ is small and, consequently, the uncertainty in the associated parameters $\omega$ and $\phi$ is high. Regardless of temperature, model selection appears to prefer a vanishing value of $D$ for some replicates. The reason is likely that the estimates of $D$ correspond to Peclet numbers in the range from Pe $\approx 200$ to Pe $\approx 1000$, which lies close to the detection threshold; see S1H Appendix.

The frequency of speed oscillations appears linked with the population doubling time. In fact, the oscillations for 22°C to 37°C align well when time is rescaled according to the duration of a cell cycle, see Fig 5c. The speed consistently attains a minimum after one doubling time $\tau_2 = \log(2)/\Lambda$ when the next replication cycle starts and the number of active forks is thus increased. As comparisons, when plotting the oscillations against absolute time since replication initiation (Fig 5d) or against replication progress (Fig 5e) the alignment is substantially worse.

The speed oscillations were first observed and quantified using a model in which speed is modulated in space, rather than in time [21]. Our general theory permits to analytically solve also a spatially modulated speed model in the small-noise limit. The resulting parameter values well match those of Ref. [21], and are consistent with our temporally modulated model, see S1G Appendix for details. The model with temporal speed oscillations yields a better fit for the majority of samples, consistently with the idea that the speed variations are linked with the doubling time. The improvement in likelihood is, however, small (Fig G2b in S1G Appendix).

## Conclusions

In this paper, we introduced a general theory that connects the DNA replication program with the abundance of DNA fragments that one should expect in an asynchronously growing population of cells. Our theory builds on previous approaches [3, 15, 21, 22, 25] and has the advantage of being based on a minimal set of realistic assumptions and allowing for stochastic replication programs. As we have demonstrated, these key properties make our theory applicable to a broad range of organisms, from bacteria to eukaryotes.

We have used our approach to estimate the origins location and intensities in budding yeast from the DNA abundance distribution measured in [43]. Our approach is based on seminal work by Bechhoefer and coworkers [1, 38], that we extended to asynchronously growing populations. A previous study [3] also attempted at extending the approach from [1, 38] to asynchronously growing budding yeast. Our results differ from those of Ref. [3] in two different aspects. First, in fitting the model to the data, Ref. [3] used prior knowledge of the origin locations. Instead, our method was able to directly infer these coordinates, without requiring any species-specific information other than the unannotated reference genome and the DNA abundance distribution. In this respect, our approach is much simpler than existing methods to map origins of replication in budding yeast [44–47]. Second, Ref. [3] assumed as a working hypothesis a uniform distribution for the distribution $P(\tau)$. In contrast, we have shown that the distribution $P(\tau)$ should be exponential under very general conditions.

In our eukaryotic model, we assumed for simplicity that replisome speed is constant; that origins are placed at well defined sites; and that they fire at an origin-dependent rate that is constant in time. The last assumption, in particular, is a drastic approximation, since origin firing rates in yeast are known to be markedly time-dependent [8, 48]. Relaxing these assumptions constitutes an important challenge for future research and will permit to recover origin timing behaviour, beside locations, and thus provide a more complete picture of the replication program.

In any case, despite these simplifying assumptions, our algorithm successfully recovers the locations of the majority of known origins in budding yeast, with an accuracy on the order of

kilobases. The accuracy can likely be further increased by exploiting advances in sequencing technology, in particular increased sequencing depth and read lengths, and by further improving the optimization algorithm. Our results demonstrate that the combination of deep sequencing of asynchronous populations and our inference approach provides a cost-effective way of discovering the replication origins of any single-cellular eukaryotic species which can be cultured and sequenced.

In the case of bacterial DNA replication, we proposed a model in which the replisome speed is modulated in time and replisomes can stochastically stall. In our model, stochastic stalling is described in terms of a biased diffusion process, see Eq (11). This idea is reminiscent of the dynamics of RNA polymerases, where such a mechanism has been experimentally tested [49]. It will be interesting in the future to quantitatively test whether this mechanism is consistent with the single-molecule dynamics of DNA polymerases.

We solved our bacterial replication model exactly and fitted its prediction against sequencing data of *E.coli* growing at different temperatures [21]. The fits show that the period of speed oscillations matches the population doubling time, or equivalently the time interval between consecutive origin firing. Our model with time-periodic speed variations fits the data slightly better than the one with space-periodic variations as postulated in [21]. Taken together, these observations support that the causes of oscillations are linked with the cell cycle, or alternatively with the origin firing rate. A possible candidate would be competition among multiple forks on the same genome [21]. Ref. [21] observed a correlation between speed oscillations and genome-wide variations in mutation rate as reported in [50]. Our results suggest that both variations are caused by a time-dependent mechanism. Further work is needed to elucidate the possible causal link between these two phenomena.

At variance with Ref. [21], the approach introduced in this paper leads to an analytical expression for the DNA abundance distribution, which considerably simplifies the inference procedure and provides additional physical insight.

Our approach reveals that the replisome speed fluctuations in *E.coli* are rather small. On the one hand, this observation confirms that simpler approaches that neglect stochasticity [22, 23] provide reliable results, at least in the case of wild type *E.coli*. On the other hand, speed fluctuations, albeit small, provides important information about the uncertainty of the replisome meeting point. In *E.coli*, the Tus-*Ter* system is know to set bounds on the region in which replisomes can meet [51–53], thereby likely affecting this accuracy. Our model predicts that, in wild type *E.coli* under laboratory conditions, the replisome diffusivity is so small that the Tus-*Ter* system is barely exercised and has therefore a negligible effect on the DNA abundance distribution, see S1F Appendix. It will be interesting for future studies to apply our approach to mutant strains, to see whether they are characterized by a different degree of uncertainty.

## Supporting information

**S1 Appendix. Appendices A-H containing detailed derivations and algorithms.**
(PDF)

## Acknowledgments

We thank S. Hauf, C. Plessy, and Y. Yokobayashi for fruitful discussions. We thank A. Alsina, J. Bechoefer, N. Rhind, P. Sartori, and A. Sassi for feedback on a preliminary manuscript.

## Author Contributions

**Conceptualization:** Florian G. Pflug, Simone Pigolotti.

**Formal analysis:** Florian G. Pflug, Deepak Bhat, Simone Pigolotti.

**Funding acquisition:** Simone Pigolotti.

**Investigation:** Florian G. Pflug, Deepak Bhat, Simone Pigolotti.

**Methodology:** Florian G. Pflug, Simone Pigolotti.

**Software:** Florian G. Pflug, Deepak Bhat.

**Supervision:** Simone Pigolotti.

**Writing – original draft:** Florian G. Pflug, Deepak Bhat, Simone Pigolotti.

**Writing – review & editing:** Florian G. Pflug, Deepak Bhat, Simone Pigolotti.

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
