## [Decision Letter · Decision Letter 0]

13 Nov 2023

Dear Pigolotti,

Thank you very much for submitting your manuscript "Genome replication in asynchronously growing microbial populations" for consideration at PLOS Computational Biology. As with all papers reviewed by the journal, your manuscript was reviewed by members of the editorial board and by several independent reviewers. The reviewers appreciated the attention to an important topic. Based on the reviews, we are likely to accept this manuscript for publication, providing that you modify the manuscript according to the review recommendations.

Sincerely,

Oleg A Igoshin

Academic Editor

PLOS Computational Biology

Jason Haugh

Section Editor

PLOS Computational Biology

Reviewer's Responses to Questions

**Comments to the Authors:**

Reviewer #1: Referee report D-23-1621: "Genome replication in asynchronously growing microbial populations", by Plug et al.

In the current manuscript, the authors present a new method to predict

the abundance of DNA fragments in asynchronously growing

cultures. They use this approach to successfully infer the origins of

replication in budding yeast, showing that this method can be used to

discover replication origins from sequencing data. The application of

their scheme to DNA replication in E. coli shows that it predicts

temporal variations in the replication speed with a period that scales

with the cell doubling time, suggesting that these variations are

related to the cell cycle.

The generic nature of the scheme makes the study very

interesting. Moreover, the manuscript is well written and the results

are clearly presented. I can therefore recommend publication of this

manuscript in PLCB after the authors have satisfactorily addressed the

following questions and comments:

- For yeast, the authors clearly state that they assume that the

firing rate is constant in time. But what do they assume for

E. coli? I presume they assume all origins fire once per cell

cycle. But do they include stochasticity in the precise moment of

firing? And can they use their method to infer (correlated versus

uncorrelated) variations in the times at which the respective

origins fire? If so, what would their analysis predict for the time

between the firing of the first and last origin? Skarstadt et

al. have measured this to be on the order of 3 - 4 minutes

(Skarstsadt et al., EMBO J., 1986; see also Berger and Ten Wolde,

PRXL, 2023). If this inference is possible, I presume the authors

could also convolve the variations in the timing with the evolution

of the volume with time, to predict variations in the initiation

volume. Would that agree with the variance in the initiation mass as

measured experimentally (Wallden et al., Cell, 2016; Si et al.,

Curr. Biol. 2019; Boesen et al. bioRxiv:2022.09.08.507175 (2022)?

- After an origin has fired, there is typically a refractory time that

prevents these origins from rapid refiring. How does that affect the

results? Certainly for E. coli at high growth rates, with cell

doubling times of 20 - 30 minutes, the refractory time, of order 10

minutes, is relatively large.

- A related question: Which assumptions could be relaxed or

introduced, to improve the agreement with experimental data? For

example, would imposing a refractory time or stochasticity in the

firing moment improve the agreement with experimental data?

- Could the authors expand their discussion of the origin of the

variations in replication speed. In their previous manuscript

(eLife, 2022), they suggested it is related to variations in the

mutation rate. Does the current analysis still support that idea? I

presume the observation that the period of the speed variations

scales with the doubling time, rather than spatial position,

provides evidence against this hypothesis? A more detailed

discussion would be appreciated.

Minor comments:

- It would help to explain f(x,τ) in the caption of Fig. 1.

- It would help to explain ψ(x,τ) in words, the first time it is introduced, i.e. above Eq. (3).

Reviewer #2: The authors propose a general theory for extracting information about the DNA replication program from DNA abundance data. The theory is based on a few fundamental assumptions that are valid in most situations, and some of them can potentially be relaxed. The theory begins with a stochastic setting but also generates analytical results for the deterministic limit. One of the advantages of this theory is its applicability to both eukaryotic and prokaryotic cells. It is demonstrated that in eukaryotic cells, the theory can be used to estimate the number and locations of origins, while in prokaryotic cells, it can be used to calculate variations in the replication speed of the two replisomes. Predictions from the theory align well with the data in both cases. Due to its generality, this theory has the potential to be exploited in future studies on DNA replication in various species.

Major comments:

About E. coli. replication, the paper assumed that the replication terminates once the two replisomes meet each other. However, replication termination should be determined by terC where both replisomes stop. How would the result be different if they made this more realistic assumption?

Minor comments:

1. Line 184: what is the indication of a 7 kb binning length scale? Are there references to this length scale?

2. In Eq. (11), the speed modulation function h(τ) is assumed to be the same for the two replisomes.

However, since the sequences that the two replisomes replicate are not symmetric, h(τ) can be different. In this case, do the two replisomes’ speeds strongly correlate with each other? Also, in Conclusions, the authors speculate that the competition between the two replisomes can be a source of speed fluctuation. How would that affect h(τ) in the model?

3. There are similarities between the E. coli. results in this paper and their 2022 Elife paper (https://doi.org/10.7554/eLife.75884), including the model itself, part of the figures, and the main conclusions. It would be worth emphasizing the new information this paper provides.

Reviewer #3: This paper presents an original approach to the study of DNA replication, with applications to both E. coli and budding yeast. The focus is on the inference of origin locations as determined from populations of cells that grow asynchronously, using experimental data based on deep sequencing (counting local DNA content). The fact that the data comes from asynchronous cells is significant because such experiments are not only easier to do than ones that attempt to synchronize cell populations, they also avoid various experimental artifacts that arise as a result of direct synchronization. Synchronization can also be done by sorting (FACS and related techniques) but is also only approximate.

The approach taken here generalizes previous work on analysis of DNA replication in bacteria that accounts for the possibility of re-replication and, most impressively, presents a unified framework that is demonstrated to be relevant for both bacterial and eukaryotic replication data. I found Eq. (9) a particularly pretty result. And the derivation of Eq. (6) is much more straightforward than the one previously given in Ref. [22].

Although I have a number of main and minor comments, I am overall very impressed by this paper and think it makes a significant contribution. Many of the comments are either to clarify what has been done or to press a bit harder for what might be easy generalizations.

Main Comments:

1) The title is a bit off, in that ~1/3 of the text concerns eukaryotic replication and indeed, the authors claim that the theory they develop “equally applies to bacteria and eukaryotes” (L49). A title that included both cases would more accurately reflect the contents of the paper. I think this is important because this paper is perhaps unique in advancing a description of DNA replication kinetics that is demonstrated to be relevant for both prokaryotic and eukaryotic organisms.

2) Please comment more about the way stalling is modeled. I get that Eq. 11 is a convenient way for having a process with stalls, but it gives a temporal stochastic mechanism whereas previous discussion of stalls tends to view it as location specific. See, for example, the discussion and related references in Ref. [22]. Does this difference matter? Also, please elaborate on the properties of the stalls. What is the distribution of stall durations? Using the Péclet number to assess the significance of stalls seems plausible, but can one justify it more precisely? Finally, does this discussion have anything to do with the v = v(x) or v(t) discussion in the paper / SI?

3) How sensitive is the model and analysis to the assumption of perfect asynchrony? How could one assess that asynchrony experimentally? The authors refer to this issue briefly but somewhat opaquely in Appendix A, after (A2). I didn’t really follow their comments there. For example, can one relate the growth function to a distribution of cell-cycle “phases”? And, again, can one determine from the data the form of that function?

Minor Comments:

Author summary: infer the origins of replication in budding yeast [there are more than one]

Line 13: “some origin sites may not be activated at all” makes it sound like this is a rare occurrence. But in budding yeast, more than half of the origins are not activated in a given cell cycle.

Lines 90–95: the definition of mathcal{P}(x) is not as clear as it might be. In particular, what do the authors mean by “randomly chosen genome”? As usually defined, genome includes all locations x, so the sentence makes it sound like mathcal{P}(x) = 1, by definition. Since that isn’t what is meant, the sentences should be written more carefully, and perhaps have a sketch to illustrate.

Line 122: can not → cannot

Line 179: The authors mention that they infer 234 origins. This would be a good place to mention that K=354 were found in the original study and to briefly discuss. (The number is buried in the caption to Fig. 3, but it seems natural to discuss here.)

Fig. 3 and surrounding discussion: Please give more details on the correlation measurement (define the quantity presented). Also, please be more precise in what was done for the shifted and shuffled surrogate data sets.

For the Langevin equations (11), please discuss more why it is appropriate to include the modulation in both the velocity and the noise strength.

Line 232: smoothen → smooth

Fig. 5 caption: Bacterial model fitted → Bacterial model fit

Line 296: In discussing the generalization to variable initiation rates, in addition to Ref. [10], it would be good to refer to A. Goldar et al, “Universal Temporal Profiles of Replication Origin Activation in Eukaryotes”, PLoS ONE 4, e5899 (2009).

Lines 311–312: “fork firing”. Do the authors mean origin firing? (In the usual language, the origins fire, creating two forks that then move or “progress”.)

**Have the authors made all data and (if applicable) computational code underlying the findings in their manuscript fully available?**

Reviewer #1: Yes

Reviewer #2: Yes

Reviewer #3: None

PLOS authors have the option to publish the peer review history of their article (what does this mean?). If published, this will include your full peer review and any attached files.

Reviewer #1: No

Reviewer #2: No

Reviewer #3: No

Figure Files:

Data Requirements:

Reproducibility:

References:

---

## [Editor Report · Decision Letter 1]

11 Dec 2023

Dear Pigolotti,

We are pleased to inform you that your manuscript 'Genome replication in asynchronously growing microbial populations' has been provisionally accepted for publication in PLOS Computational Biology.

Best regards,

Oleg A Igoshin

Academic Editor

PLOS Computational Biology

Jason Haugh

Section Editor

PLOS Computational Biology

---

## [Editor Report · Acceptance letter]

27 Dec 2023

PCOMPBIOL-D-23-01621R1 

Genome replication in asynchronously growing microbial populations

Dear Dr Pigolotti,

I am pleased to inform you that your manuscript has been formally accepted for publication in PLOS Computational Biology. Your manuscript is now with our production department and you will be notified of the publication date in due course.

With kind regards,

Zsofi Zombor
